# Psychological Risk Factors that Predict Social Networking and Internet Addiction in Adolescents

**DOI:** 10.3390/ijerph17124598

**Published:** 2020-06-26

**Authors:** Montserrat Peris, Usue de la Barrera, Konstanze Schoeps, Inmaculada Montoya-Castilla

**Affiliations:** 1Department of Personality, Evaluation and Psychological Treatments, University of the Basque Country, 20018 San Sebastian, Spain; montserrat.peris@ehu.eus; 2Department of Personality, Assessment and Psychological Treatment, Faculty of Psychology, University of Valencia, 46010 Valencia, Spain; usue.barrera@uv.es; 3Department of Psychology, Faculty of Health Sciences, European University of Valencia, 46010 Valencia, Spain; konstanze.schoeps@universidadeuropea.es

**Keywords:** adolescents, internet addiction, social networking, body self-esteem, personality traits, fsQCA models

## Abstract

Adolescents’ addictive use of social media and the internet is an increasing concern among parents, teachers, researchers and society. The purpose was to examine the contribution of body self-esteem, personality traits, and demographic factors in the prediction of adolescents’ addictive use of social media and the internet. The participants were 447 Spanish adolescents aged 13−16 years (*M* = 14.90, *SD* = 0.81, 56.2% women). We measured gender, age, body self-esteem (body satisfaction and physical attractiveness), personality traits (extraversion, neuroticism, disinhibition and narcissism) and social networking and internet addiction (internet addiction symptoms, social media use, geek behaviour, and nomophobia). The effects of gender, age, body self-esteem and personality on the different dimensions of internet addiction were estimated, conducting hierarchical linear multiple regression analysis and a fuzzy-set qualitative comparative analysis (fsQCA). The results evidenced different pathways explaining four types of adolescents’ internet addiction: gender and disinhibition were the most relevant predictors of addiction symptoms; gender combined with physical attractiveness best explained social media use; narcissism and neuroticism appear to be the most relevant predictors of geek behaviour; and narcissism was the variable that best explained nomophobia. Furthermore, the advantages and differences between both methodologies (regressions vs. QCA) were discussed.

## 1. Introduction

### 1.1. Risk of Internet Addiction and Social Media Use

The use of social networking sites and the internet has grown in popularity over the last few decades and new technological tools such as smartphones may have become indispensable today [1,2]. Prevalence rates vary considerably in internet addiction research. Across Europe, recent studies reported prevalence ranging between 4.4% to 13.5% for pathological internet use and between 14.3% and 54.9% for problematic internet use [3]. In Spain, the prevalence of problematic internet users has been estimated to be between 18.5% and 4.9% of pathological internet users [4]. Adolescence is an especially vulnerable period of change and teenagers face the risk of suffering symptoms of addiction as a result of their daily social network use [2]. Traditionally, internet addiction has been analysed in a generalised and global way, however, recent studies suggest the relevance of investigating each subtype of addiction in each particular population [5]. For instance, Peris et al. [6] consider four types of risk behaviour regarding social networking and internet addiction in adolescents on a subclinical level: internet addiction symptoms, social media use, geek behaviour, and nomophobia.

Internet addiction symptoms express the users’ urge to continue being connected despite the desire to stop, experiencing unpleasant emotions when they do not succeed. Adolescents, who perceive the constant need to be online, usually present an increased use of social networking sites and the internet. In extreme cases, such behaviour may produce psychological problems that traditionally correspond to substance-related addictions [2]. Addiction symptoms would include sleep disturbance, angry or agitated reaction when forced to disconnect, and loosing track of time while online [6]. Personal characteristics such as gender and age appear to play an essential role in internet addiction [7], however, the results are not conclusive. On the one hand, recent studies report that girls experience higher amount of internet addiction symptoms than boys [4,8,9]. On the other hand, some other studies reveal that boys are more susceptible to addictive online behaviour than girls [10]. With regard to age difference, the results are also inconclusive. On the one hand, some studies report a progressive increase in addiction to the internet with age, enhancing also the negative consequences [11], while other studies report a progressive decrease [12]. On the other hand, there are studies that do not observe any age differences at all [9]. Hence, research investigating the influence of gender and age on internet addiction during adolescence seems warranted.

The aspect of social media use refers to a virtual way of relating with peers, reducing face-to-face interaction. Social networking sites have become a new environment of group socialization for adolescents. Such virtual environments enable the users to create public profiles in an interactive space where they communicate with friends but also with strangers, people they never have contact with outside the network [13]. Excessive use of social networking sites, however, can have negative implications such as negative mood states, concentration problems, and less interest in spending time with friends and family [14]. In relation to gender, it appears that women tend to use smartphones primarily for communication and social media such as Instagram and Facebook [15,16]. On the contrary, other studies have reported that boys are more likely to present problematic social media use than girls [17]. With regard to age difference, studies reveal that the risk and frequency of addictive use of social networking sites decrease as age increases [18].

Geek behaviour refers to a certain habit, which is commonly ascribed to an intense level of interest in the technological field. The term *geek* is widely used among young people in the digital era that is why there is still little literature about the phenomenon. In this study, the term geek behaviour refers to the quality of young people who are passionate about information and communication technologies (ICT), including online social networking. In addition, they usually prefer interacting with their peers through the internet, given the easy access to websites where you can make friends online or look for an online date [19]. Three characteristics have been identified: relating through online games, searching for similar interest groups and/or online eroticism [20]. With regard to gender, research consistently shows that boys engage more frequently in compulsive gaming and gambling [15,21]. Although boys show typically more geek behaviour, some studies have evidenced such behaviour is increasing among girls, too [22]. Furthermore, research stresses that being an adolescent is a well-established risk factor for internet gaming addiction [23].

Nomophobia is conceptualized as the intense fear of being disconnected from smartphone communication on a nonclinical level [24]. Messaging apps have increased nomophobia, especially in teenagers, the age group most affected by this problem. Currently, the accelerating development of smartphones devices and the fact that they are easy to carry around all the time, means that mobile phones are replacing the internet as a primary addictive source [25]. In relation to gender, studies suggest that girls use their smartphones more often than boys and that their use is, therefore, more problematic [16]. Hence, addictive behaviour in boys is associated with gaming apps, while in girls smartphone use is more related to online communication and social interaction [15]. With regard to age, adolescents have been targeted as a group at risk for smartphone addiction [26] and a recent meta-analysis suggests that one in four adolescents presents problematic smartphone use [27].

Social cognitive theory recognizes “personal factors” as potential predictors of internet addiction disorder and nomophobia, considering gender, age, ethnicity or beliefs as the most relevant. However, the contribution of other psychological factors such as self-esteem and personality traits should be considered in order to better understand of such risk behaviour online.

### 1.2. Body Self-Esteem and the Relationship with Problematic Social Media and Internet Use

Body self-esteem is another psychological variables that has been positively associated with the social networking addiction [28]. The internet is the virtual mirror of the current body image by exposing a manipulated online version with an increased physical and erotic appeal [29]. Frequent social media users have greater body dissatisfaction because they have internalized the model of a thin and long-legged body [30]. On the one hand, physical attractiveness -the emotional dimension of body self-esteem- may correlate positively with social networking and internet addiction [31], while body satisfaction -the cognitive dimension of body self-esteem- may correlate negatively. Recent studies reveal that adolescents who are dissatisfied with their bodies tend to selectively disclose their very positive and attractive features (e.g., their most attractive photographs). Such behaviour may work as a self-preservation strategy that helps to restore their self-confidence and receive positive feedback from their peers, which in turn generates positive feelings about themselves [32]. These idealized self-preservation motivations are positively associated with social networking addiction [33].

On the other hand, body satisfaction seems to be related to social desirability, thus, peers play a decisive role in the matter of public online appearance. Peris et al. [34] reported that an increasing number of adolescents edit their pictures before posting them on social networking sites to improve their image according to how they wish to see themselves and be seen by others. Following this line of research, studies indicate a negative association between adolescent body satisfaction and the use of social networking sites. According to Liu et al. [35], adolescents who are unsatisfied with their body, experience low self-esteem in real life and, therefore, more often seek positive responses through social media. In turn, positive responses from online social interactions may reinforce young people’s use of smartphones, enhancing the risk of developing smartphone addiction. If adolescents would receive more positive responses and support in real life, they might refrain from frequent online interactions, even if they have low body satisfaction.

### 1.3. Personality Traits and the Association with Social Networking and Internet Addiction

In addition to self-esteem, personality has an important bearing on explaining individual differences in problematic internet use [33,36]. One of the most studied personality dimension related to internet addiction is extraversion, which refers to people’s varying tendency to be friendly, sociable, and talk active. Studies suggest that there is a positive relationship between being extroverted and an increased risk of internet addiction [37]. Regarding the internet and social media use, they are both strongly and positively related to extraversion [38]. Extraverted individuals use social networking apps to engage in social interaction and they may attain more social resources online. In relation to the geek behaviour, Dieris-Hirche et al. [39] reported that gamers with a problematic internet use scored lower on extraversion and higher on neuroticism than those with non-problematic behaviour. Furthermore, as smartphones allow for the permanent connection demanded by extroverted people, extraversion has been identified as a predictor variable of problematic smartphone use [40]. However, some recent studies did not confirm such relationship [41]. These discrepancies may suggest the importance of a differential conceptualization of social networking and internet addiction in relation to different kinds of risk behaviour.

Another main personality factor is neuroticism, which refers to individual differences in emotional stability and psychological adjustment. Neuroticism has been shown a positive association with problematic internet use [42]. Studies suggest that introverted adolescents show increased online activity when compared with emotionally stable users, which may enhance the risk for internet addiction [27]. Regarding the social aspect of internet use, neurotic adolescents tend to show a more passive behaviour on social networking sites such as posting comments and Likes on other people’s profiles, which may represent a way to seek social relationships [33,36,43]. This association can be interpreted in terms of their difficulties to pursue social relationships in real life and poor coping skills with emotional situations [42]. Research on the relationship between neuroticism and geek behaviour is limited. Furthermore, neurotics tend to show an increased or problematic use of online games [39]. With regard to nomophobia, neuroticism appears to predict obsessive smartphone use [44].

Besides extraversion and neuroticism, disinhibition is an other relevant personality domain, which is associated with increased internet addiction symptoms [45]. Research indicates that problematic internet and social media use is linked to different aspects of disinhibited behaviour such as poor self-control, impulsivity and sensation seeking. Disinhibited personality is common among teenagers, but usually decreases with age from adolescence to adulthood, supporting the hypothesis of normative psychological maturation. Apparently, for disinhibited adolescents the internet may represent a useful tool for satisfying their elevated urge of sensation seeking [45] by allowing relief from unpleasant feelings through quick online behaviour without thinking much of the consequences (e.g., by quickly engaging in anonymous virtual relationships) [31]. In view of the relevance of this personality domain related to addictive internet behaviour, it appears negligent that only few studies have included disinhibition as a potential predictor of social networking and internet addiction.

Moreover, narcissism has been related to problematic internet use in adolescence [46]. Narcissism may be understood as a dynamic self-regulation system related to grandiosity, self-absorption and a constant need for external approval. There is evidence to suggest that higher levels of narcissism are linked to an increased risk of social networking and internet addiction [28,37]. Social media use appear to fulfil two basic social needs that may explain the problematic internet use in young people with narcissistic personality: the need for self-presentations through social online profiles featured by pictures, status updates, personal notes, etc.; and the need to belong to a social group seeking repeated approval from their peers [47]. Furthermore, narcissistic teenagers tend to use social media such as Facebook more often because such platforms enable users to create visible profiles, that allow idealized self-promotion in a virtual environment [37]. In addition to problematic social media use, narcissism is strongly associated with smartphone addiction [48].

### 1.4. Rationale for the Study

Drawing from previous studies, there are psychological risk factors of significant relevance during adolescence such as gender and age, body self-esteem and personality traits, including extraversion, neuroticism, disinhibition and narcissism that predict internet-related addictions [16,21,31,37]. Most studies use a very limited conceptualization of social networking and internet addiction rather than broadening their view and focus on specific addictive behaviours (e.g., [5,45]). To our knowledge, there are no studies that have investigated the combined effects of body self-esteem and personality traits in order to determine different types of risk behaviour regarding the social networking and internet addiction in nonclinical adolescents, involving internet addiction symptoms, social media use, geek behaviour, and nomophobia. Additionally, the role of demographic factors such as gender and age have been extensively studied in relation to adolescent problematic internet use, however with mixed findings [8,9,10,16]. In addition, most research on social networking and internet addiction has used methodology based on linear regression models (e.g., [1,49]). These models are focused on the individual prediction and do not consider the possibility of different pathways that would lead to the same result [50,51]. In contrast, fuzzy-set qualitative comparative analysis (fsQCA) is a methodology that allows a more in-depth analysis of how a set of causal conditions contribute to a hypothesised outcome [52,53]. QCA is based on the assumption that such outcome depends on a combination of different factors rather than on individual levels of those factors [54,55]. In the field of psychological research, limited studies have used this technique despite it’s considerable potential [56].

### 1.5. Purpose of the Study

The present study aimed to estimate the combined contribution of body self-esteem (body satisfaction and physical attractiveness), personality traits (extraversion, neuroticism, disinhibition and narcissism), and demographic factors (gender and age) in the prediction of four types of adolescent’s social networking and internet addiction (internet addiction symptoms, social media use, geek behaviour, and nomophobia). Based on the reviewed research, we hypothesized as follows (1) the primary risk factors that predict internet addiction symptoms will be gender and age (girls of older age), low body satisfaction, and high levels of physical attractiveness, neuroticism, extraversion, disinhibition and narcissism; (2) the potential risk factor of social media use will be gender and age (girls of younger age), low body satisfaction, but high levels of physical attractiveness, neuroticism, extraversion, disinhibition and narcissism; (3) the significant risk factors that predict geek behaviour will be gender (boys), low body satisfaction and extraversion, but high physical attractiveness neuroticism, disinhibition and narcissism; and (4) the main risk factors of nomophobia will be gender (girls), low body satisfaction, but high levels of physical attractiveness, neuroticism, extraversion, disinhibition and narcissism.

## 2. Materials and Methods

### 2.1. Participants

The present research involved 447 adolescents aged 13−16 years (*M* = 14.90, *SD* = 0.81). The gender distribution of the sample was equitable (women: *n* = 251; 56.2%). The participants were students from public (*n* = 201; 45%) and private (*n* = 246; 55%) high schools in the Northern Regions of Spain. The following inclusion criteria applied: (a) school board gave their permission to collaborate with the research group; (b) students were not older than 16 years; (c) parents or guardians were asked to sign a written consent to allow adolescents’ participation in the research. The simple random probability sampling method has been employed.

### 2.2. Variables and Instruments

#### 2.2.1. Demographic Variables

Personal data referring to the students’ gender, age and high school were collected administrating an ad hoc questionnaire.

#### 2.2.2. Social Networking and Internet Addiction

The social networking and internet addiction was assessed using the Scale of risk of addiction to social media and the internet for adolescents (ERA-RSI) [6]. The scale is composed of 29 items divided into four subscales: internet addiction symptoms, social media use, geek behaviour, and nomophobia. The internet addiction symptoms subscale assesses behaviours of addiction to non-toxic substances (e.g., “I have been losing sleep over social media and watching online shows”; 9 items). The social media use subscale assesses adolescent “online socialization” behaviours (e.g., “I check my friends’ profiles”; 8 items). The geek behaviour subscale includes aspects such as joining special interest groups, playing online and role-playing games or having sexual encounters (e.g., “I spend time on social media and the Internet to play online and/or role-playing games”; 6 items). The nomophobia subscale is related to feelings of anxiety and control when using a mobile phone (e.g., “I have a smartphone and I start feeling anxious or distressed when people don’t answer immediately to my messages”; 6 items). Participants were asked to rate the items on a 4-point Likert scale (1 = *never or hardly never*; 4 = *many times or almost always*). The reliability indexes were adequate in this sample for the global scale of addiction (α = 0.90) and all subscales: internet addiction symptoms (α = 0.84), social media use (α = 0.83), geek behaviour (α = 0.69), nomophobia (α = 0.80).

#### 2.2.3. Body Self-Esteem

The body self-esteem was measured with the Body Self-esteem Scale (BSS) [57]. This scale is composed of 26 items and participants evaluate the degree of satisfaction with each part of their body. The first 20 assess body satisfaction, the cognitive dimension of body self-esteem, and are grouped into four body areas: face (e.g., “Are you satisfied with your eyes/mouth?”), upper torso (e.g., “Are you satisfied with your breasts/pectorals?”), lower torso (e.g., “Are you satisfied with your butt?”); and anthropometry (e.g., “Are you satisfied with your height/size?”). The score ranges on a 10-point Likert scale (1 = *very dissatisfied*, 10 = *very satisfied*). Physical attractiveness, the emotional dimension of body self-esteem, is assessed through 6 items and six aspects are evaluated: physically interesting, socially charming, sexy, attractive, sensual and erotic (e.g., “To what extent do you consider yourself a physically attractive person?”). The score ranges on a 10-point Likert scale (1 = *not attractive at all*, 10 = *very attractive*). This scale shows good psychometric properties in the studied sample (Body satisfaction α = 0.93; Physical attractiveness α = 0.93; Body Self-esteem α = 0.95).

#### 2.2.4. Personality Factors

The extraversion and neuroticism dimensions were assessed using NEO Five Factory inventory (NEO-FFI) [58], the brief version of NEO-PI-R. The scale is composed of 60 items distributed equally on five factors (12 items in each factor). Two subscales have been used in the present study: neuroticism (e.g., “I am quite emotionally stable”–inversed item) and extraversion (e.g., “I like having a lot of people around me”). They were chosen for two reasons. First, they are the ones that accumulate the most research with regard to social media and internet use, and second, because it was necessary to reduce the application time due to the fatigue and tiredness. The scale uses a 5-point Likert scale (0 = *strongly disagree*; 4 = *strongly agree*). Both factors have shown suitable reliability in this sample (extraversion: α= 0.81; neuroticism: α= 0.79).

The disinhibition was evaluated using the Sensation Seeking Scale (SSS-Q) [59], Spanish adaptation [60]. The scale is composed of 40 items, which give rise to 4 subscales (10 items each). We selected the subscale of disinhibition for the aim of our study, given that its content is most relevant for risk behaviour and addiction. The participants provide information about their own disinhibition behaviours (e.g., “I like wild parties without limits“). The adolescents can answer the items *affirmatively* (1) or *negatively* (0). The scale has shown good reliability in this sample (α = 0.64).

The narcissism was evaluated using the Narcissistic Personality Inventory (NPI) [61], Spanish adaptation (NP-15) [62]. The scale is composed of 15 items and the narcissism includes facets such as a need for recognition, an exalted and distorted image or a feeling of special status (e.g., “It’s very important that others pay attention to me and admire what I do”). The scale uses a 6-point Likert scale (1 = *absolutely false*; 6 = *absolutely true*). The scale has shown adequate reliability in this sample (α = 0.81).

### 2.3. Procedure

The present research was approved by the Ethics Commission for Research with Human Beings (CEISH) of the University of the Basque Country Euskal Herriko Unibersitatea (UPV/EHU), with the registration number CEISH/136/2012/PERIS HERNANDEZ. The declaration of the file was in the Basque Agency of Data Protection with the registration number 2080310015-INA0004. The study applied the ethical principles of the Declaration of Helsinki, the requirements established by the Ethics Commission for Human Research of the University of the Basque Country (UPV) and the State Government, as well as the deontological regulations of the Official College of Psychologists for experimentation on human beings.

First, the headmasters and psychologists of each school were contacted and information about the study and a copy of the research project were provided. Once they accepted to participate, the parents of the adolescents were contacted. An information letter and informed consent was sent to their homes. The informed consent was signed and returned to the school by the adolescent’s parents or legal guardians in order to participate. The adolescents were informed about the questionnaires and confidentiality, and they signed the informed consent. The data collection took place in the school classrooms during the tutorial hour. The questionnaires were administered by evaluators with a Degree in Psychology and lasted approximately 40 min.

### 2.4. Data Analysis

Firstly, descriptive analysis followed by bivariate correlations and linear regression analyses were conducted using the statistical package IBM SPSS V.25 for Windows (IBM Corporation, Foster City, CA, USA). A three-stage hierarchical multiple regression analyses was performed to examine the predictive power of demographic variables (age and gender), body self-esteem (body satisfaction and physical attractiveness), and personality factors (neuroticism, extraversion, disinhibition and narcissism) on social networking and internet addiction. A total of four regression models were carried out, one for each dimension of addiction and predictors were entered in three stages: (a) Gender and age, (b) body self-esteem and (c) personality factors.

Secondly, we performed fuzzy-set qualitative comparative analysis (fsQCA) using the Fs/QCA 3.0 software (University of California, Irvine, CA, USA). Prior to conducting the analysis, we estimated the calibration scores, transformed raw data responses into fuzzy-set responses, and removed missing data. In order to obtain the constructs (variables) and increase the variability, the items of each scale were multiplied [63]. Following the multiplication of the items, the extraversion and neuroticism scales were divided by one hundred to avoid excessively large numbers that the program could not handle. The rest of the variables remained undivided. In addition, each variable was then recalibrated in three categories: percentile 10 (low levels or condition is absent), percentile 50 (intermediate level, condition is neither absent nor present) and percentile 90 (high levels or condition is present) [64]. All scores must range between 0 and 1. Thus, gender and age scores were calibrated manually. Age scores were coded according at four points equidistant between 0 and 1, gender scores were recoded with 0 for boys and 1 for girls. We conducted descriptive analyses with the transformed scores. Finally, necessary and sufficient conditions analyses estimated the combined influence of the demographic variables, body self-esteem and personality factors on high levels of internet addiction symptoms, social media use, geek behaviour and nomophobia. There is no theoretical number of combinations that produces the outcome.

## 3. Results

### 3.1. Descriptive Analysis and Relationships Between Variables Studied

Descriptive statistics (means and standard deviations) and correlations between the study variables are displayed in Table 1. Results indicated that age was significantly and in a negative way related to social media use, nomophobia, neuroticism and extraversion, while the associations with disinhibition and narcissism are positive. In general, personality was positively and significant related to social networking and internet addiction. Specifically, disinhibition was associated with the four dimensions of addiction; neuroticism and extraversion were related to internet addiction symptoms, social media use and nomophobia, but not with geek behaviour; and there were a positive association between narcissism and internet addiction symptoms, geek behaviour and nomophobia, but there are not with social media use. With regard to body self-esteem, physical attractiveness was significantly and positively correlated with the four dimensions of addiction to the internet and online social networking, whereas body satisfaction only correlated to internet addiction symptoms and the relationship was negative. As regards the relationship between personality and self-esteem, neuroticism was negatively and significant related to body satisfaction and attraction, whereas extraversion, disinhibition and narcissism were related positively, although there was not relationship between disinhibition and body satisfaction.

### 3.2. Demographic and Psychological Predictors of Social Networking and Internet Addiction

Predictive analysis of social networking and internet addiction was conducted with a three-step hierarchical multiple regressions (Table 2).

Regarding the first prediction model, three sets of variables were established, which explained 37% of the variance of adolescents’ internet addiction symptoms. In the first step, which included demographic variables, specifically gender and age, explained 9% of the variance. The second next step, which included two dimensions of body self-esteem, accounted for an additional 10% of explained variance. The third and final step included the four dimensions of personality explained an additional 19% of variance. The final regression model indicates that gender, body satisfaction, physical attractiveness, neuroticism, extraversion, disinhibition and narcissism are significant predictors of emotional symptoms.

The second prediction model consisted of three sets of variables, which explained 35% of the variance of social media use. In the first step, gender and age were included and together they explained 23% of the variance. In the second step, the two dimensions of body self-esteem were entered and accounted for a significant increase of 5% of the variance. In the final step, the four personality dimensions were included, which accounted for an additional 10% of variance. In this final model, gender, age, physical attractiveness, neuroticism, extraversion, and disinhibition significantly predicted social media use.

With regard to the third prediction model, the overall regression model was significant but only explained 5% of geek behaviour. Following the same *modus operandi,* demographic variables were included in step 1 explaining 1% of the variance, followed by both dimensions of body self-esteem in step 2 increasing the explained variance by 3% and finally all personality four dimensions were entered in step accounting for an additional 3% of the variance. The resulting model suggested that only disinhibition and narcissism significantly predict geek behaviour.

The forth the prediction model was established in three steps and explained 21% of the variance of nomophobia. Firstly, gender and age were entered in the first step and explained 8% of the variance. Secondly, body satisfaction and physical attractiveness were included in the second step elevating the explained variance by 8%. Thirdly, neuroticism, extraversion, disinhibition and narcissism were added in the third step and together explained an additional 9 % of the variance. In this overall model, gender, age, physical attractiveness, neuroticism, disinhibition and narcissism significantly predicted nomophobia.

### 3.3. Combined Contribution of Body Self-Esteem, Personality Traits and Personal Predictors of Social Networking and Internet Addiction

The descriptive statistics of the variables under study and the calibration values were calculated first (Table 3). The dimensions of social networking and internet addiction based on personality, body self-esteem and personal factors were then examined by fuzzy-set QCA. The necessary and sufficient conditions were estimated. On the one hand, the analysis of necessary conditions or variables allows us to determine whether there are any variables that are always required to be present for the prediction of the outcome (in our study, high levels of addiction). For a condition/variable to be necessary, the consistency score must be above 0.90 [51]. The results showed that none of the variables studied have to be considered necessary condition because their consistencies scores were all below 0.90. On the other hand, the sufficient conditions refer to those variables that predict the outcome, but the prediction may be possible without them. All logically possible combinations of the causal conditions are captures in the truth table together with the result of each setting [50]. The analysis provides three types of solutions: a parsimonious one (less restrictive), an intermediate one and a complex one (most restrictive). The literature recommends, therefore, focusing on the intermediate solution [51], which is presented in this study.

In the social networking and internet addiction, the combination of conditions resulting in high levels of internet addiction symptoms, social media use, geek behaviour and nomophobia were analysed (Table 4). The solutions were found to be adequate, considering that a fsQCA model is acceptable when consistency is above 0.70 [50]. The main three pathways for high levels of all four dimensions of social networking and internet addiction are shown in Table 4.

The solution showed eight pathways, which explained 46% of high levels of internet addiction symptoms. The first pathway was the result of the combined contribution of a high gender score (girl), high age, high disinhibition and high narcissism. The second pathway to predict high levels of internet addiction symptoms was the combined contribution of a high gender score (girl), high neuroticism, high extraversion and high disinhibition and the third was the result of the interaction of a high gender score (girl), high neuroticism, high extraversion, high disinhibition and high narcissism.

The solution showed 12 pathways, which explained 56% of high levels of social media use. The first pathway was the result of the combined contribution of high gender score (girl), high neuroticism and high physical attractiveness. The second pathway was the combined contribution of high gender score (girl), low neuroticism and high disinhibition. The third pathway was the result of the interaction of a high gender score (girl), high age, high extraversion and high physical attractiveness.

The solution showed eight pathways, which explained 33% of high levels of geek behaviour. The first combination was the result of the interaction of a high gender score (girl), high age, high neuroticism, high extraversion, high disinhibition and high narcissism. The second pathway was the result of the combined contribution of high gender score (girl), high age, high neuroticism, high extraversion, high narcissism, high body satisfaction and high physical attractiveness, and the third pathway was the result of the interaction of a low gender score (boy), high neuroticism, high narcissism and low physical attractiveness.

Finally, the solution showed 11 pathways, which explained 41% of high levels of nomophobia. The first pathway was the result of the interaction of a high gender score (girl), high extraversion and high narcissism. The second pathway was the result of the interaction of a high neuroticism, high extraversion, high disinhibition, high narcissism, low body satisfaction and high physical attractiveness. The third pathway predicting a high nomophobia was the result of the combined contribution of a low age, high neuroticism, high disinhibition, high narcissism and low body satisfaction.

The solution showed 12 pathways which explained 56% of high levels of social media use. The first pathway was the result of the combined contribution of high gender score (girl), high neuroticism and high physical attractiveness. The second pathway was the combined contribution of high gender score (girl), low neuroticism and high disinhibition. The third pathway was the result of the interaction of a high gender score (girl), high age, high extraversion and high physical attractiveness.

The solution showed eight pathways which explained 33% of high levels of geek behaviour. The first combination was the result of the interaction of a high gender score (girl), high age, high neuroticism, high extraversion, high disinhibition and high narcissism. The second pathway was the result of the combined contribution of high gender score (girl), high age, high neuroticism, high extraversion, high narcissism, high body satisfaction and high physical attractiveness, and the third pathway was the result of the interaction of a low gender score (boy), high neuroticism, high narcissism and low physical attractiveness.

Finally, the solution showed 11 pathways which explained 41% of high levels of nomophobia. The first pathway was the result of the interaction of a high gender score (girl), high extraversion and high narcissism. The second pathway was the result of the interaction of a high neuroticism, high extraversion, high disinhibition, high narcissism, low body satisfaction and high physical attractiveness. The third pathway predicting a high nomophobia was the result of the combined contribution of a low age, high neuroticism, high disinhibition, high narcissism and low body satisfaction.

## 4. Discussion

Although research on social networking and internet addiction is on the rise, a vast majority of investigations have conceptualized internet addictive as a one-dimensional construct, focusing on problematic or compulsive use rather than specific risk behaviour. We aimed to estimate the combined contribution of body self-esteem (body satisfaction and physical attractiveness), personality traits (extraversion, neuroticism, disinhibition and narcissism), and personal factors (gender and age) in the prediction of four types of adolescents’ social networking and internet addiction, involving internet addiction symptoms, social media use, geek behaviour, and nomophobia. Several studies have investigated the relations between addiction to the internet and social networking and several of the variables included within this study [65]; however, the addiction literature using a novel methodological approach by comparing linear regression models with fsQCA models in the prediction of adolescents’ internet addiction is sparse. Thus, this study makes an innovative contribution to the addiction literate by evaluating a more comprehensive model of addiction to internet and social media, examine different pathways of variables representing important biologically based personality traits, variables of body self-esteem, which are of special relevance during adolescence and taking into account gender and age differences. The findings extend our understanding of the psychological risk factors of social networking and internet addiction among nonclinical adolescents.

### 4.1. Risk Factors of Addiction Symptoms

The first hypothesis has been supported by the results from the hierarchical regression and fsQCA models, which showed a significant influence of gender, body self-esteem and personality traits on internet addiction symptoms. Both methodologies match the result that gender and disinhibition were the most relevant predictors of internet addiction symptoms. The regression model suggests that adolescents, more girls than boys, who are more disinhibited, neurotic, narcissistic and extraverted, and experience lower body satisfaction but higher physical attractiveness, present more internet addiction symptoms. The results from fsQCA suggested that the combination of gender (being a girl) and high levels of disinhibition were the most significant predictors of internet addiction symptoms. The two latter in combination with high levels of narcissism, neuroticism and extraversion also predicted an increase in internet addiction symptoms, but to a lesser extent. In contrast to the regression models, results from QCA suggests that body self-esteem does not seem to be an important predictor of internet addiction symptoms, since they have not been included in the three most significant pathways. These findings are in line with recent research, indicating that girls tend to present more addiction symptoms in comparison with boys [8,16]. However, these results are inconsistent with other studies that suggest that boys are more likely to show addictive internet behavior [10]. Evidentially, boys and girls may be at risk of potential internet addiction, but probably for different reasons. While girls are usually more interested in the social interactions over the internet, boys, contrarily, use the internet primarily for online gaming [3]. Furthermore, disinhibition has been the primary psychological risk factor for adolescents’ internet addiction symptoms in our study, highlighting the role of impulsive behaviour as a manifestation of poor self-regulating skills in relation to problematic internet use during adolescence [45]. In general, personality factors appear to be more important than body self-esteem in the prediction of internet addiction, matching research finding that have pointed out the important of personality traits in explaining individual differences in symptoms of internet addiction [33,36].

### 4.2. Risk Factors of Social Media Use

Overall, the second hypothesis is also supported by the results obtained in our study, suggesting that the combination of demographic and psychological variables influence adolescents’ problematic social media use. Both regression and QCA models indicate that gender combined with physical attractiveness seem to be the most relevant predictors of social media use. On the one hand, results from the hierarchical regression models revealed that girls more than boys, younger adolescents more than the older ones, with a physically attractive body image and a disinhibited, neurotic and extraverted personality, tend to use social media more often. On the other hand, fsQCA models suggest that there are different pathways to predict the problematic social media use. One of the main pathways shows that girls with higher physical attractiveness in combination with higher neuroticism report more excessive social media use. Another pathway suggests that girls with high levels of disinhibition in combination with low levels of neuroticism also show increased social internet use. A third pathway combines girls of older age with high physical attractiveness and high extraversion, in the prediction of addictive use of social networking sites. These differences may suggest that the combination of personality factors and body self-esteem may vary producing different patterns of risk behaviour. Stressing the role of neuroticism is critical, given that both high and low levels of emotional stability in combination with other psychological risk factors predict social media use. Our findings are relatively conforming with previous studies, which have provided hard evidence that girls use social media more often than boys [8,15]. In a recent study, Escario and Wilkinson (2020) [21] have provided evidence that men and women relate differently to the internet. For instance, while women tend to participate more in social activities, using chat rooms, sending e-mails and visiting social networking sites such as Facebook or Instagram, men spend more time on online games, online gambling, and visit pornographic sites more frequently. Our findings also corroborate previous studies that have reported a positive relationship between adolescents’ body image and the use of social media [31]. In fact, teenage girls feel more pressured to present themselves with an idealized physical attractiveness by editing their self-portraits before sharing them in social media, which in turn generates positive feelings by receiving increased external approval, hence, higher risk of social networking addiction [2,7]. With regard to personality factors, our findings were consistent with previous studies, showing that disinhibition, neuroticism and extraversion were positively associated with addictive use of social networking sites [42,45].

### 4.3. Risk Factors of Geek Behaviour

Additionally, the third hypothesis has been supported by our results in terms of adolescents’ geek behaviour, however the predictive capacity of demographic and psychological factors is lower than for the other types, irrespective of the methodology used. The personality factors narcissism and neuroticism appear to be the most relevant predictor of geek behaviour. The results from regression models indicate that high levels of narcissism and disinhibition predict geek behaviour, explaining only a small amount of variance. In fsQCA analysis, the model included more factors than in the other models in order to improve the prediction outcome, producing pathways with many different combinations of variables. For instance, the main pathway suggests that girls of older age with high levels of narcissism, neuroticism, extraversion and disinhibition tend to be geekier. Furthermore, the second pathway shows that girls of older age, who are neurotic, extraverted and narcissistic and experience higher levels of body satisfaction and physical attractiveness, also present more geek behaviour. Lastly, the third pathway indicates that boys with high levels of narcissism and neuroticism combined with an unattractive body image also predict greater geek behaviour. Both boys and girls, especially the older ones, appear to score high on geek behaviour, depending on the combination of several psychological risk factors. In fact, a single dimension of personality traits or body self-esteem is not sufficient in order to explain the individual difference in this type of internet addiction. These unexpected results are, however, compatible with some previous research, which suggests that both boys and girls may present geek behaviour [8]. Traditionally, geek behaviour related to excessive online gaming and gambling has been related to the male gender [15,21,65]. However, girls seem also to be at risk of such behaviour especially related to online eroticism, which is a novel insight regarding current literature. Another important finding of this study is that narcissism and neuroticism are the main predictors of geek behaviour. There is some evidence, that neurotic and introverted adolescents show increased online gaming behaviour [39]. Nevertheless, the role of narcissism in the prediction of geek behaviour has not been established in prior research, thus, our findings may give rise to a new line of research. The strong link between narcissism and geek behaviour may be explained by the “compensatory perspective”, which suggest that online gaming might fulfil a compensatory purpose for narcissistic individuals with emotional dysregulation [66]. Regarding the role of body self-esteem, our results are less conclusive. Previous research have associated gamers’ body dissatisfaction or negative body attitudes with the exposure to ideal video game bodies [67].

### 4.4. Risk Factors of Nomophobia

With regard to nomophobia, the results of this study confirm the forth hypothesis, demonstrating a significant impact of demographic factors, body self-esteem and personality traits. However, it should be highlighted that narcissism has been the most influential factor in the prediction of nomophobia in both methodologies. Based on the results of the hierarchical regressions, more girls than boys, younger adolescents more than the older ones, with an attractive body image and a primarily narcissistic, disinhibited and neurotic personality, report higher levels of nomophobia. Similarly, gender and age appear in the primary combinations of fsQCA that predict nomophobia. In addition, low levels of body satisfaction in combination of high levels of physical attractiveness also contribute to the prediction of nomophobia. Even though narcissism is the most significant predictor, high levels of neuroticism, extraversion and disinhibition also appear in two of the three main combinations. In line with previous studies, our findings provide further evidence that personality in general and narcissism specifically are significant predictors of adolescents’ nomophobia [28,43]. Results from a recent studies indicate that the link between narcissism and smartphone distress may be explained by increased attention-seeking [48]. One of the characteristics of nomophobia is the constant checking for instant notifications, which may act as reward, but on the other hand, increases the level of anxiety and distress [68]. Hence, the need for affirmation by narcissistic adolescents may encourage greater dependence on social media that manifests itself in increased anxiety or signs of nomophobia. Furthermore, our findings match previous research on the relative effect of demographic variables on nomophobia. On the one hand, girls appear to feel more anxious and insecure regarding their smartphones, and on the other hand, nomophobia levels increase along with age [49]. Finally, our findings suggest different patterns for cognitive versus emotional components of body self-esteem in the prediction of nomophobia [69]. It can be argued that adolescents, who are less satisfied with their body and therefore promote a more attractive image of themselves in social media and internet, tend to feel more anxious and insecure when they are disconnected from their smartphones, similar as the experience of social rejection.

In conclusion, the results of both regression and QCA analyses suggest that demographic variables, body self-esteem and personality traits significantly influence adolescent social networking and internet addiction. On the one hand, girls with a disinhibited personality, in addition to high levels of neuroticism, narcissism and extraversion but to a lesser extend, tend to present more symptoms of internet addiction. Meanwhile, girls, who feel physically attractive and describe themselves as disinhibited and extraverted tend to use social media more often. Neuroticism also seems to be an influence with different patterns: low levels of neuroticism in combination with high levels of disinhibition, or high levels of neuroticism in combination with greater physical attractiveness, predict increased social media use. On the other hand, adolescents, both girls and boys, with high levels of narcissism and neuroticism show greater geek behaviour. The influence of body self-esteem is unclear. Finally, adolescents, mostly girls of younger age, who are less satisfied with their body but show greater physical attractiveness, with high levels of narcissism, and also high levels of neuroticism, extraversion and disinhibition but to a lesser extend, are more likely to present more signs of nomophobia.

### 4.5. Strengths, Limitations and Further Research

The strengths of this present research are both theoretical and methodological. On the one hand, our study focuses on internet addiction as multidimensional concept rather than a one-dimensional approach. The instrument of measurement of addiction to social media and the internet has been specifically designed for adolescents and contemplates the four different types of addictive online behaviour: (a) symptoms that excessive internet use entails; (b) the use of social networking sites for virtual interactions with peers, due to the fact that at this age peer relationship are more and more important; (c) geek behaviour, which is characterized by an intense level of interest in online gaming and online sexuality typical of adolescence; and (d) finally nomophobia defined as the problematic use of the mobile phone, which starts at early adolescence. On the other hand, the methodological approach is the second strengths of this study. If both methodologies are compared in the prediction of social networking and Internet addiction in adolescence, fsQCA models cover a greater number of factors than regression models. In addition, fsQCA allows for multiple pathways to be estimated by combining the predictors in different ways, depending on the relationships between variables. For these reasons, fsQCA methodology may be considered a complementary analytical approach to traditional regression models.

It is necessary to stress the limitations of the study. With regard to sampling method and size, the sample size was appropriate for conducting multiple regressions and fsQCA, however probability sampling does not guarantee the generalization of the results obtained. It would be recommendable in future research to carry out a cluster-stratified random sampling that includes adolescents from all over the country. With regard to data collection, we believe that self-reports completed by the adolescents were appropriate for the purpose of the study. Future research, however, may use mixed methods (qualitative and quantitative data), multiple reports from parents and peers, which would provide more in-depth information about adolescents’ addiction. Finally, one of the main limitations of fsQCA is the limited number of predictor variables that can be included in the analyses. While in regression models an increase in sample size allows an increase in the number of predictors, in fsQCA the maximum number of conditions is invariable.

This research contributes to the study of potential risk factors of social networking and internet addiction in adolescence, regardless of the limitations that has been considered. Moreover, this study offers a more comprehensive conceptualization of internet addiction by identifying four different types of addictive online behaviour in young people and describing the different pathways of variables representing important biologically based personality traits, variables of body self-esteem, which are of special relevance during adolescence and taking into account gender and age differences. Our findings extend previous addiction literature, providing an in-depth analysis of several combinations of psychological and demographic variables that may increase the risk of potential behavioural addictions.

## 5. Conclusions

The implications of this study are both theoretical and practical. Overall, this study makes a unique contribution to the literature on addictive use of social networking sites and broadens the way for further research that may provide additional evidence towards other adolescent-relevant variables. Such research would provide health professionals with relevant information about individual differences in the four types of social networking and internet addiction, and therefore, enriching their professional experience when working with affected adolescents. Furthermore, from a practical point of view, the findings of our study may help psychotherapists to make decisions on who to prioritize a certain intervention and treatment approach based on the relevance of the reported risk factors such as personality and gender. Such stable characteristics may be detected quickly in adolescents in order to identify those who are most at risk and thus start a preventive intervention at an early stage. Thus, the study of youth internet addiction is essential in order to improve prevention and early intervention. Finally, this study has identified important risk factors that underlie the psychological mechanisms of social networking and internet addiction in adolescents. Stressing the benefits of identifying adolescent internet addiction symptoms, problematic social media use, geek behaviour and nomophobia, through a specific instrument that allows for a multidimensional assessment.

## Figures and Tables

**Table 1 ijerph-17-04598-t001:** Bivariate correlations between all variables studied.

	1	2	3	4	5	6	7	8	9	10	11
1. Age	−										
2. IAS	−0.07	−									
3. SMU	−0.17 ***	0.56 ***	−								
4. GB	−0.02	0.29 ***	0.30 ***	−							
5. NP	−0.16 **	0.61 ***	0.55 ***	0.32 ***	−						
6. NT	−0.12 *	0.36 ***	0.29 ***	0.03	0.25 ***	−					
7. EX	−0.11 *	0.12 ***	0.20 ***	0.04	0.09 ***	−0.19 ***	−				
8. DI	0.16 ***	0.40 ***	0.23 ***	0.16 ***	0.23 ***	0.12 **	0.20 ***	−			
9. NA	0.11 *	0.21 ***	0.02	0.18 ***	0.18 ***	0.06	−0.11 *	0.20 ***	−		
10. BS	−0.01	−0.15 ***	−0.02	0.06	−0.03	−0.32 ***	0.20 ***	−0.03	0.10 *	−	
11. PA	0.06	0.13 **	0.11 **	0.15 ***	0.13 **	−0.20 ***	0.25 ***	0.22 ***	0.26 ***	0.66 ***	−
***M***	14.90	19.05	21.17	9.31	12.62	32.89	46.27	14.73	40.45	6.58	6.18
***SD***	0.81	5.77	4.86	2.77	4.21	7.31	6.33	2.13	10.13	1.32	1.67

Note. IAS = Internet addiction symptoms. SMU = Social media use. GB = Geek behaviour. NP = Nomophobia. NT = Neuroticism. EX = Extraversion. DI = Disinhibition. NA = Narcissism. BS = Body satisfaction. PA = Physical attractiveness. *M* = mean. *SD* = standard deviation. * *p* ≤ 0.05. ** *p* ≤ 0.01. *** *p* ≤ 0.001.

**Table 2 ijerph-17-04598-t002:** Results of hierarchical multiple regression analyses.

Variable	Internet Addiction Symptoms				Social Media Use				Geek Behaviour				Nomophobia			
	*ΔR^2^*	*ΔF*	β	*t*	*ΔR^2^*	*ΔF*	β	*t*	*ΔR^2^*	*ΔF*	β	*t*	*ΔR^2^*	*ΔF*	β	*t*
Step 1	0.09	21.93 ***			0.23	64.84 ***			0.01	1.62			0.08	20.43 ***		
Gender			0.27	6.50 ***			0.45	10.67 ***			−0.06	−1.08			0.26	5.51 ***
Age			−0.06	−1.45			−0.08	−2.07 *			−0.07	−1.34			−0.14	−3.12 **
Step 2	0.10	28.34 ***			0.05	12.47 ***			0.03	5.77 **			0.05	13.08 ***		
Body satisfaction			−0.18	−3.34 ***			0.04	0.76			−0.04	−0.54			−0.05	−0.87
Physical attractiveness			0.21	3.89 ***			0.11	1.99 *			0.12	1.76			0.16	2.59 **
Step 3	0.19	33.64 ***			0.10	16.51 ***			0.03	3.52 **			0.09	12.66 ***		
Neuroticism			0.23	5.44 ***			0.18	4.16 ***			0.03	0.59			0.15	3.20 ***
Extraversion			0.09	2.11 *			0.15	3.44 ***			0.01	0.18			0.05	1.15
Disinhibition			0.30	7.06 ***			0.20	4.63 ***			0.10	2.02 *			0.17	3.56 ***
Narcissism			0.16	3.93 ***			0.04	1.00			0.13	2.57 *			0.17	3.68 ***
*Durbin-Watson*	1.27				1.26				1.61				1.33			
*R^2^*	0.37 ***				0.35 ***				0.05 **				0.21 ***			

Note. *ΔR^2^ =* change in *R^2^; ΔF =* change in *F;* ß = regression coefficient; *t* = value of *t*-test statistic; * *p* ≤ 0.05. ** *p* ≤ 0.01. *** *p* ≤ 0.001.

**Table 3 ijerph-17-04598-t003:** Descriptive statistics and calibration scores.

		IAS	SMU	GB	NP	NT	EX	DI	NA	BS	PA
Mean		5164.83	6055.23	32.66	181.55	13973.16	210162.93	69.35	746999.25	2261.04	113609.19
Standard deviation		19216.76	10541.97	93.71	328.93	54894.57	336946.74	112.60	4809591.82	1705.54	156534.70
Minimum		1.00	1.00	1.00	1.00	0.01	0.25	1.00	0.02	21.96	1.00
Maximum		262144.00	65536.00	972.00	2304.00	703125.00	2441406.25	1024.00	84375000.00	10000.00	1000000.00
Calibration scores											
Percentile	10	5.60	70.40	1.00	1.00	11.52	6635.52	4.00	6.48	439.84	1896.00
	50	384.00	1944.00	6.00	48.00	518.40	78643.20	32.00	2764.80	1837.68	63504.00
	90	10368.00	15552.00	72.00	524.80	22413.31	562500.00	128.00	813957.12	4724.43	290304.00

Note. IAS = Internet addiction symptoms. SMU = Social media use. GB = Geek behaviour. NP = Nomophobia. NT = Neuroticism. EX = Extraversion. DI = Disinhibition. NA = Narcissism. BS = Body satisfaction. PA = Physical attractiveness.

**Table 4 ijerph-17-04598-t004:** Three main pathways for the high levels of social networking and internet addiction.

*Frequency Cutoff: 1*	High Internet Addiction Symptoms			High Social Media Use			High Geek Behaviour			High Nomophobia		
	Consistency Cutoff: 0.90			Consistency Cutoff: 0.90			Consistency Cutoff: 0.90			Consistency Cutoff: 0.90		
	**1**	**2**	**3**	**1**	**2**	**3**	**1**	**2**	**3**	**1**	**2**	**3**
Gender	●	●	●	●	●	●	●	●	○	●		
Age	●					●	●	●				○
Body satisfaction								●			○	○
Physical attractiveness				●		●		●	○		●	
Neuroticism		●	●	●	○		●	●	●		●	●
Extraversion		●				●	●	●		●	●	
Disinhibition	●	●	●		●		●				●	●
Narcissism	●		●				●	●	●	●	●	●
Raw Coverage	0.24	0.23	0.23	0.29	0.28	0.26	0.13	0.12	0.12	0.21	0.17	0.17
Unique Coverage	0.012	0.044	0.010	0.017	0.020	0.004	0.026	0.015	0.008	0.042	0.028	0.016
Consistency	0.88	0.90	0.91	0.91	0.88	0.91	0.89	0.89	0.89	0.85	0.94	0.90
Overall Solution Coverage			0.46			0.56			0.33			0.41
Overall Solution Consistency			0.86			0.83			0.85			0.85

Note. ● = presence of condition/high levels, ○ = absence of condition/low levels. Gender: ● = girls; ○ = boys. All sufficient conditions are adequate.

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
