# Peer review of "Psychological Risk Factors that Predict Social Networking and Internet Addiction in Adolescents"

_ijerph, 2020, doi:10.3390/ijerph17124598_

Round 1

Reviewer 1 Report

At first glance I have found this paper as outstanding due to the importance of the topic and the use of professional research tool. Moreover, there are pointed out practical implication of the study,  that results could be used by psychotherapists to prioritize their intervention. A very interesting methodological approach was applied by comparison of fuzzy-set qualitative comparative analysis (fsQCA) and hierarchical regression models. This is the added value of this paper, but at the same time makes it difficult to pick up.

I have the following comments in order of importance:

  • The greatest objections are raised by the mean values of the scales used, which do not correspond to the values from other studies provided in the cited literature. Such high values seem impossible with a given number of items and Likert scale. In addition, it is difficult to compare the results obtained with other works with such a scoring system. Please describe better the scoring system and rationale of using it.
  • There is also a contradiction in the concept of the study, which may be due to its dual nature (research and methodological). The fsQCA method seems to be an exploratory one, focused on discovering hidden and unknown behavioral patterns. On the other hand, the authors formulate hypotheses about the expected results (lines 227+). It would be better to pose research questions instead of hypotheses. It will cause small changes in discussion.
  • Please change the title of table 4, to Three most important…… You have described 8/12/8/11 conditions. It took me some time to link it with the contents of table 4
  • Please give the theoretical number of conditions in the method section.
  • Please add all found conditions as supplementary material, because you suggested practical implications
  • The discussion should have subchapters related to research questions (hypotheses); comparison of two methods and limitations & strengths.
  • The limitations of the study should be extended, especially those coming from the fsQCA method. The sample size is not large and maybe not all conditions (patterns) were discovered. Please discuss the sample size in the context of number of conditions, and some weaknesses of this method. percentiles were calculated from the relative small sample. 
  • The order of tables is not clear. Why descriptive results are presented after the regression model.
  • I encourage Authors to shorten the text and reduce number off references, to make the text more reader-friendly. 
  • Please remove Spanish letters (or printing marks?), eg.2.2.3. subchapter about body image
  • I am not sure if Fs/QCA 3.0 software was only used. What about regression model?

Author Response

Reviewer 1

At first glance I have found this paper as outstanding due to the importance of the topic and the use of professional research tool. Moreover, there are pointed out practical implication of the study, that results could be used by psychotherapists to prioritize their intervention. A very interesting methodological approach was applied by comparison of fuzzy-set qualitative comparative analysis (fsQCA) and hierarchical regression models. This is the added value of this paper, but at the same time makes it difficult to pick up.

I have the following comments in order of importance:

Point 1: The greatest objections are raised by the mean values of the scales used, which do not correspond to the values from other studies provided in the cited literature. Such high values seem impossible with a given number of items and Likert scale. In addition, it is difficult to compare the results obtained with other works with such a scoring system. Please describe better the scoring system and rationale of using it.

Response 1:

We appreciate the reviewer’s comment and incorporated the mean values and standard deviations using the original scale correction and scoring system in the table 1 (p.8, line 336). Correlation and regression analyses have been conducted using the traditional scoring system proposed by the original authors of the measurements used in this study. However, for fsQCA analyses the raw scores are transformed into calibration values, which are displayed in Table 3. FsQCA methodology needs a higher variability to perform the transformation of raw scores to calibration values between 0 and 1. For this reason, instead of calculating the scales using means or sum, items are multiplied. This is the reason why the calibration values (Table 3) mentioned by the reviewer are so high.

Furthermore, we thought it would be useful for the reviewer to better understand this novel methodological approach and, therefore, we provided a more detailed description of Qualitative Comparative Analysis (QCA):

Qualitative comparative analysis is an analytic technique that builds on set theory and that allows for an in depth analysis of how causal conditions contribute to an outcome. QCA assume that the influence of particular attributes (variables) on a specific outcome depends on how the attributes combine, rather than on the levels of the individual attributes.

In QCA methodology usually includes crisp sets and fuzzy sets based on Boolean algebra. The crisp set refers to binary (not necessarily nominal) cases (or variables) and, therefore, only allow two possible values: 0.0 when the condition is absent in the case (full non-membership); and 1.0 when the condition is present in the case (full membership). For example, gender is a crisp variable with two possible values, 0.0 for male and 1.0 for female. The fuzzy set is used fo ordinal, interval or ratio cases (variables) and includes the possibility of intermediate values between 0.0 and 1.0. The raw data have to be transformed into calibration values to assume values that range between 0 and 1.

In a first step, due to the fact that fuzzy-set needs a great variability of raw scores, the total scores are calculates by multiplying the items that belong to each scale, instead of calculating the sum or the average. Occasionally, the values are excessively high and the FS/QCA program is not able to handle them, so the total scores are divided by 100. In a second step, three cut-off points (axes of comparison) for each variable are estimated using the 10th, 50th and 90th percentile ranks (PR): 10th PR means the condition is absent in the case (low levels of the variable), 50th PR means the condition is neither absent nor present in the case (intermediate levels of the variable), and 90th PR means the condition is present in the case (high levels of the variable). Following these cutt-off points, all the raw data is recoded to assume values that range between 0 and 1.

Once all the variables have been recoded and values range between 0 and 1, the Fuzzy-set Qualitative Comparative Analysis (fsQCA) can be performed to analyze both the crisp sets and the fuzzy sets.

Firstly, the necessary conditions or variables have to be estimated. A necessary condition refers to a variable that must be present for a certain outcome to occur. Thus, for a condition to be considered necessary, the consistency must be greater than .90 in the sufficiency analyses. In the present study, no variable has obtained a consistency greater than .90, hence, there is no necessary condition.

Secondly, the sufficient conditions or variable are calculated. A sufficient condition refers to a variable that can produce the outcome by itself. For this purpose, a truth table algorithm transforms the fuzzy-set membership values into a truth table that order cases over the logically possible combinations of causal conditions and assigns the outcome to each configuration. The frequency cut off is set to 1 and the consistency cut off is set to .90, to obtain the highest coverage without losing consistency.

The program Fs/QCA 3.0 performs the analysis and offers three types of solutions: the parsimonious (least restrictive), the intermediate and the complex (most restrictive). According to Ragin (2008), the intermediate solution should be used for interpreting of the results. Regarding the intermediate solution, the number of pathways is shown that lead to high levels of the analyzed condition or variable. Finally, the three main pathways are reported in the text, this is, those with the highest raw coverage.

For more details please consult the following book: Schneider, C.Q., & Wagemann, C. (2012). Set-theoretic methods for the social sciences: a guide to qualitative comparative analysis. Cambridge: Cambridge University Press.

Point 2: There is also a contradiction in the concept of the study, which may be due to its dual nature (research and methodological). The fsQCA method seems to be an exploratory one, focused on discovering hidden and unknown behavioral patterns. On the other hand, the authors formulate hypotheses about the expected results (lines 227+). It would be better to pose research questions instead of hypotheses. It will cause small changes in discussion.

Response 2:

We appreciate the reviewer’s comment and concern regarding the formulation of hypotheses.

However, an fsQCA analysis is perfectly compatible with the purpose of this study and the formulation of hypotheses.

On the one hand, qualitative comparative analysis is an analytic technique that builds on set theory and that allows for an in depth analysis of how causal conditions contribute to an outcome. QCA assume that the influence of particular attributes (variables) on a specific outcome depends on how the attributes combine, rather than on the levels of the individual attributes.

On the other hand, the procedure of fsQCA analyses is similar to the one of hierarchical regressions, where you enter different blocks of variables according to their importance in the prediction of a certain outcome based on findings from previous literature. Thus, in fsQCA the conditions or variables are entered in the analyses, indicating weather high or low levels of each independent variable will predict high levels of the outcome variable, also based on findings from previous literature. More specifically, if low levels of the independent variable are expected to predict high levels of the outcome, the predictor is entered as "low". The analysis would than indicate whether actually “low levels” or rather "high levels" of the predictor variable influence in high levels of the outcome or it does not influence the pathways at all. The FS/QCA software then estimates the combined contribution or pathways that produce the expected outcome to contrast the proposed hypothesis.

For these reasons, fsQCA is compatible with formulating hypotheses. However, it should be taken into account that the hypotheses are based on studies that have mostly used traditional regression models rather than combined contribution of the variables studied.

In addition, there are recent studies that compare regression models and QCA models, which have also formulated hypotheses:

Villanueva, L., Valero-Moreno, S., Cuervo, K., & Prado-Gascó, V. (2019). Sociodemographic variables, risk factors, and protective factors contributing to youth recidivism. Psicothema, 31(2), 128-133. doi: 10.7334/psicothema2018.257

Navarro-Mateu, D., Alonso-Larza, L., Gómez-Domínguez, M. T., Prado-Gascó, V., & Valero-Moreno, S. (2020). I’m Not Good for Anything and That’s Why I’m Stressed: Analysis of the Effect of Self-Efficacy and Emotional Intelligence on Student Stress Using SEM and QCA. Frontiers in Psychology, 11, 295. doi:10.3389/fpsyg.2020.00295

Point 3: Please change the title of table 4, to Three most important……You have described 8/12/8/11 conditions. It took me some time to link it with the contents of table 4

Response 3:

Following the reviewer’s suggestion we changed thee title of the table 4 (p.12, line 426) to “Three main pathways for the high levels of social networking and internet addiction”.

Furthermore, the pathways 8/12/8/11 correspond to the combinations of conditions, which has been obtained for each type of addiction. In Table 4, the three main pathways in the prediction of social networking and internet addiction are displayed.

More specifically, table 4 shows the main results of the intermediate solutions in the analysis of high levels of internet addiction symptoms, social media use, geek behavior and nomophobia through personal variables, body self-esteem and personality traits. The three main paths are shown with the numbers 1, 2 and 3 in the corresponding columns. The black circles represents full-membership, meaning that the condition is present or high levels of the variable, while the white circles represent non full-membership, meaning that the condition is not present or low levels of the variable. If there is no circle, than his variable does not appear in this combination that produces the outcome. Overall Solution Coverage refers to the total coverage value of the intermediate solution paths and Overall Solution Consistency refers to consistency, which is usually considered acceptable above .70.

Point 4: Please give the theoretical number of conditions in the method section.

Response 4:

We incorporated the reviewer’s suggestion and addressed the theoretical number of combination (pathways) in the method section (p. 7, line 318).

There is no theoretical or predetermined number of combinations that produces the outcome. All conditions (variables) are entered into the program based on the findings from previous literature. FS/QCA software then estimates the combined contribution or pathways that produce the expected outcome. These analyses are performed taking into account the Frequency cutoff and the Consistency cutoff. The results that are provided by the software indicate the different number of pathways for each type of solution.

Point 5: Please add all found conditions as supplementary material, because you suggested practical implications

Response 5:

All recent studies using the QCA methodology only include the three main pathways because of their relevance in the interpretation of the results. These main pathways are those with the highest coverage and best consistency indices. Including all combinations would imply showing almost 40 pathways, which would lead to confusing results and would make it difficult to reach operational conclusions. Please consult the following studies:

Castellano Rioja, E., Valero-Moreno, S., Giménez-Espert, M. del C., & Prado-Gascó, V. (2019). The relations of quality of life in patients with lupus erythematosus: Regression models versus qualitative comparative analysis. Journal of Advanced Nursing, 75, 1484-1492. doi:10.1111/jan.13957

Giménez-Espert, M. C., Valero-Moreno, S., & Prado-Gascó, V. (2019). Evaluation of emotional skills in nursing using regression and QCA models: A transversal study. Nurse Education Today, 74, 31–37. doi:10.1016/j.nedt.2018.11.019

Navarro-Mateu, D., Alonso-Larza, L., Gómez-Domínguez, M. T., Prado-Gascó, V., & Valero-Moreno, S. (2020). I’m Not Good for Anything and That’s Why I’m Stressed: Analysis of the Effect of Self-Efficacy and Emotional Intelligence on Student Stress Using SEM and QCA. Frontiers in Psychology, 11, 295. doi:10.3389/fpsyg.2020.00295

Villanueva, L., Valero-Moreno, S., Cuervo, K., & Prado-Gascó, V. (2019). Sociodemographic variables, risk factors, and protective factors contributing to youth recidivism. Psicothema, 31(2), 128-133. doi: 10.7334/psicothema2018.257

Point 6: The discussion should have subchapters related to research questions (hypotheses); comparison of two methods and limitations & strengths.

Response 6:

We incorporated the reviewer’s suggestion and included subchapters in the discussion section.

Point 7: The limitations of the study should be extended, especially those coming from the fsQCA method. The sample size is not large and maybe not all conditions (patterns) were discovered. Please discuss the sample size in the context of number of conditions, and some weaknesses of this method. percentiles were calculated from the relative small sample.

Response 7:

Following the reviewer’s suggestion we extended the limitations of the study, including those regarding sample size and fsQCA method (p. 16, lines 611-621).

Point 8: The order of tables is not clear. Why descriptive results are presented after the regression model.

Response 8:

We apologize for the confusion. We incorporated descriptive results in Table 1, which correspond to the mean values and standard deviations using the original scale correction and scoring system (p.8, line 336). Correlation and regression analyses have been conducted using the traditional scoring system proposed by the original authors of the measurements used in this study. However, for fsQCA analyses the raw scores are transformed into calibration values, which are displayed in Table 3.

Point 9: I encourage Authors to shorten the text and reduce number off references, to make the text more reader-friendly.

Response 9:

We appreciate the reviewer’s comment and reduced the number of references from 79 to 69. Moreover, we were able to shorten the text by over 1000 words.

Point 10: Please remove Spanish letters (or printing marks?), eg.2.2.3. subchapter about body image

Response 10:

We apologize for the mistake. We removed the Spanish question marks in the subsection 2.2.3 (p. 6, lines 243-254).

Point 11: I am not sure if Fs/QCA 3.0 software was only used. What about regression model?

Response 11:

Again, we apologize for the confusion. Descriptive analysis, bivariate correlations and linear regression analyses were conducted using the statistical package IBM SPSS V.25 for Windows (p. 7, line 295-296).

Reviewer 2 Report

The article submitted for review: Psychological risk factors that predict social networking and internet addiction in adolescents concerns a very important issue of behavioral addictions in modern youth. The authors of the article undertook - successfully, an analysis of the important in practical terms of psycho-prevention of health, the problem of predicting addictive use of social media and the Internet by adolescents. The article proposed for printing is a very good complement to the literature on the subject of the analysis of psychological aspects of the risk of social media addiction.
The article has been correctly formatted and divided into parts in accordance with the requirements of the journal, and the research methodology and statistical analysis used by the authors indicate on a mature scientific workshop.

In the opinion of reviewer, the entire article requires minor corrections, which should be introduced to the text in the following scope and parts of the article.

Introduction:
- Carefulness of the text should be increased by referring to the diagnosis of behavioral addictions (context of Internet addiction) and phobias (Nomophobia), which are presented in ICD 10 or DSM-V.

- Distinction in the text of the Aim of study part as a separate part presenting following research questions and hypotheses that the authors formulate.

Discussion:
- Clearly separate from the discussion the possible restrictions of your work (as it is now part of the discussion)

- I suggest to write a part of text Strengths, limitations and further research in order to strengthen the article

Author Response

Reviewer 2

The article submitted for review: Psychological risk factors that predict social networking and internet addiction in adolescents concerns a very important issue of behavioral addictions in modern youth. The authors of the article undertook - successfully, an analysis of the important in practical terms of psycho-prevention of health, the problem of predicting addictive use of social media and the Internet by adolescents. The article proposed for printing is a very good complement to the literature on the subject of the analysis of psychological aspects of the risk of social media addiction. The article has been correctly formatted and divided into parts in accordance with the requirements of the journal, and the research methodology and statistical analysis used by the authors indicate on a mature scientific workshop.

In the opinion of reviewer, the entire article requires minor corrections, which should be introduced to the text in the following scope and parts of the article.

Introduction:

Point 1: Carefulness of the text should be increased by referring to the diagnosis of behavioral addictions (context of Internet addiction) and phobias (Nomophobia), which are presented in ICD 10 or DSM-V.

Response 1: We incorporated the reviewer’s suggestion and reviewed the whole text for a more careful language when referring to internet addiction symptoms and nomophobia on a “sub/non-clinical level” (e.g. p. 2, line 45 and 88, p. 4, line 186)

Point 2: Distinction in the text of the Aim of study part as a separate part presenting following research questions and hypotheses that the authors formulate.

Response 2: Following the reviewer’s suggestion we included the subchapter “Aim of study” in the Introduction section.

Discussion:
Point 3: Clearly separate from the discussion the possible restrictions of your work (as it is now part of the discussion)

Response 3: We incorporated the reviewer’s suggestion and included subchapters in the discussion section in order to separate the possible limitations of our work from the rest of the discussion.

Point 4: I suggest to write a part of text Strengths, limitations and further research in order to strengthen the article

Response 4: We incorporated the reviewer’s suggestion and included subchapters “Strengths, limitations and further research” in the discussion section (p. 16, line 595).

Reviewer 3 Report

This is a nice work with clear motivation and sufficient description of method and results with thorough discussion.

The main comments are about presentation.

It would be good to have a quick introduction what are main background and what are main findings. The 3 subsections in Section 1 can be separated as a new sections 2 serving as related work and proposed hypotheses. Authors might consider to group the hypothesis with clear itemization so reader can refer to each of them in the discussion part. For same reason, in section 4, it would be good to have 3-4 subsections to highlight findings rather than very long paragraphs, then ended with a generic overall discussion.

Authors refer to literature for technical details of regression and predictor used in the analysis. But for completeness of this paper itself, it would be nicer to include a bit more details with tables or figures to explain inputs, variable values etc, esp the fuzzification and truth table, and more explanation of Table 4, would help reader to better understand the findings. A variety of presentation methods may be more attractive than plain text esp when lots of numeric data are included.

English is good enough, but worth a standard checker to massage those usage out of norms.

Author Response

Reviewer 3

This is a nice work with clear motivation and sufficient description of method and results with thorough discussion.

The main comments are about presentation.

Point 1: It would be good to have a quick introduction what are main background and what are main findings. The 3 subsections in Section 1 can be separated as a new sections 2 serving as related work and proposed hypotheses. Authors might consider to group the hypothesis with clear itemization so reader can refer to each of them in the discussion part. For same reason, in section 4, it would be good to have 3-4 subsections to highlight findings rather than very long paragraphs, then ended with a generic overall discussion.

Response 1: We appreciate the reviewer’s comment and incorporated included subchapters in the discussion section. Four sub-sections have been incorporated relating to the specific risk factors related to each of the four types of addiction, as well as sub-section outlining the strengths and limitations of the study.

Point 2: Authors refer to literature for technical details of regression and predictor used in the analysis. But for completeness of this paper itself, it would be nicer to include a bit more details with tables or figures to explain inputs, variable values etc, esp the fuzzification and truth table, and more explanation of Table 4, would help reader to better understand the findings. A variety of presentation methods may be more attractive than plain text esp when lots of numeric data are included.

Response 2: We incorporated the reviewer’s suggestion and reviewed the whole results section, removing the statistical information, which is already presented in the tables, leaving only the plain text.

Furthermore, we thought it would be useful for the reviewer to better understand this novel methodological approach and, therefore, we provided a more detailed description of Qualitative Comparative Analysis (QCA):

Qualitative comparative analysis is an analytic technique that builds on set theory and that allows for an in depth analysis of how causal conditions contribute to an outcome. QCA assume that the influence of particular attributes (variables) on a specific outcome depends on how the attributes combine, rather than on the levels of the individual attributes.

In QCA methodology usually includes crisp sets and fuzzy sets based on Boolean algebra. The crisp set refers to binary (not necessarily nominal) cases (or variables) and, therfore, only allow two possible values: 0.0 when the condition is absent in the case (full non-membership); and 1.0 when the condition is present in the case (full membership). For example, gender is a crisp variable with two possible values, 0.0 for male and 1.0 for female. The fuzzy set is used fo ordinal, interval or ratio cases (variables) and includes the possibility of intermediate values between 0.0 and 1.0. The raw data have to be transformed into calibration values to assume values that range between 0 and 1.

In a first step, due to the fact that fuzzy-set needs a great variability of raw scores, the total scores are calculates by multiplying the items that belong to each scale, instead of calculating the sum or the average. Occasionally, the values are excessively high and the FS/QCA program is not able to handle them, so the total scores are divided by 100. In a second step, three cut-off points (axes of comparison) for each variable are estimated using the 10th, 50th and 90th percentile ranks (PR): 10th PR means the condition is absent in the case (low levels of the variable), 50th PR means the condition is neither absent nor present in the case (intermediate levels of the variable), and 90th PR means the condition is present in the case (high levels of the variable). Following these cutt-off points, all the raw data is recoded to assume values that range between 0 and 1.

Once all the variables have been recoded and values range between 0 and 1, the Fuzzy-set Qualitative Comparative Analysis (fsQCA) can be performed to analyze both the crisp sets and the fuzzy sets.

Firstly, the necessary conditions or variables have to be estimated. A necessary condition refers to a variable that must be present for a certain outcome to occur. Thus, for a condition to be considered necessary, the consistency must be greater than .90 in the sufficiency analyses. In the present study, no variable has obtained a consistency greater than .90, hence, there is no necessary condition.

Secondly, the sufficient conditions or variable are calculated. A sufficient condition refers to a variable that can produce the outcome by itself. For this purpose, a truth table algorithm transforms the fuzzy-set membership values into a truth table that order cases over the logically possible combinations of causal conditions and assigns the outcome to each configuration. The frequency cut off is set to 1 and the consistency cut off is set to .90, to obtain the highest coverage without losing consistency.

The program Fs/QCA 3.0 performs the analysis and offers three types of solutions: the parsimonious (least restrictive), the intermediate and the complex (most restrictive). According to Ragin (2008), the intermediate solution should be used for interpreting of the results. Regarding the intermediate solution, the number of pathways is shown that lead to high levels of the analyzed condition or variable. Finally, the three main pathways are reported in the text, this is, those with the highest raw coverage.

Table 4 shows the main results of the intermediate solutions in the analysis of high levels of internet addiction symptoms, social media use, geek behaviour and nomophobia through personal variables, body self-esteem and personality traits. The three main paths are shown with the numbers 1, 2 and 3 in the corresponding columns. The black circles represents full-membership, meaning that the condition is present or high levels of the variable, while the white circles represent non full-membership, meaning that the condition is not present or low levels of the variable. If there is no circle, than his variable does not appear in this combination that produces the outcome. Overall Solution Coverage refers to the total coverage value of the intermediate solution paths and Overall Solution Consistency refers to consistency, which is usually considered acceptable above .70.

For more details please consult the following book: Schneider, C.Q., & Wagemann, C. (2012). Set-theoretic methods for the social sciences: a guide to qualitative comparative analysis. Cambridge: Cambridge University Press.

Point 3: English is good enough, but worth a standard checker to massage those usage out of norms.

Response 3: The manuscript has been reviewed for further mistakes typographical and grammatical issues.

Round 2

Reviewer 1 Report

The authors put a lot of work into the revision of the paper and the current version is much clearer. I see no contraindications to accept for publication.